# The association of sociodemographic characteristics and comorbidities with post-acute sequelae of SARS-CoV-2 in a Medicaid managed care population with and without HIV

Yiyi Wu[1]*, Eleni Mattas[1‡], Carey Brandenburg[2‡], Ethan Fusaris[2‡], Richard Overbey[2‡], Jerome Ernst[2‡], Mark Brennan-Ing[1]

1 Brookdale Center for Healthy Aging, at Hunter College, City University of New York, New York City, New York, United States of America, 2 Amida Care, New York City, New York, United States of America

☯ These authors contributed equally to this work.
‡ EM, CB, EF, RO and JE also contributed equally to this work.
* yyw1033@hunter.cuny.edu

## Abstract

Understanding how post-acute sequelae of SARS-CoV-2 infection (PASC) affects communities disproportionately affected by HIV is critically needed. This study aimed to identify the prevalence of PASC symptoms among Medicaid enrollees at risk for or living with HIV. Through a web survey, we received 138 valid responses from Medicaid-managed plan members who had received a COVID diagnosis. Participants' mean age was 45.4 years (SD = 11.9) and most were non-Hispanic Black (43.5%) or Hispanic (39.1%). Almost thirty-two percent reported inadequate incomes and 77.5% were HIV-positive. In the overall population, the frequently reported symptoms included neck/back/low back pain, brain fog/difficulty concentrating, bone/joint pain, muscle aches, and fatigue. Findings indicate that there is no statistically significant difference in the prevalence and intensity of PASC symptoms lasting 6 months or more between individuals living with and without HIV. Multiple regression analysis found that the number of PASC symptoms 6 months or longer was independently associated with inadequate incomes and comorbidities (cardiac problems, cancer, fibromyalgia) ($R^2$ = .34). Those with inadequate incomes and comorbidities have more numerous PASC symptoms. Implications for health care delivery and long-term COVID services will be discussed.

## Introduction

While the median recovery time for the coronavirus disease 2019 (COVID-19) is between 2 and 6 weeks, at least 10% of symptomatic patients experience lasting symptoms and complications months beyond the acute phase of the illness, known as post-acute sequelae of SARS-CoV-2 infection (PASC) or long COVID [1, 2]. The World Health Organization defines PASC

**Data Availability Statement:** The data collected for this study are not publicly available, nor are they

available on request, as it would constitute a violation of the Institutional Review Board (IRB) protocol. Additionally, as part of the informed consent agreement, participants were assured that their data would not be shared with researchers beyond our institution or used for future research purposes. We are committed to upholding these ethical standards to protect participant confidentiality and privacy.

**Funding:** This research was fully funded by an Investigator Sponsored Research Grant (CO-US-540-6404) from Gilead Sciences. The funder played a role in consulting our study design and data collection.

**Competing interests:** The authors have declared that no competing interests exist.

as the "continuation or development of new symptoms 3 months after the initial SARS-CoV-2 infection, with these symptoms lasting for at least 2 months with no other explanation [3]." Many affected by PASC experience symptoms lasting 6 months or more [4–7]. PASC outcomes can vary greatly and affect multiple organ systems [8]. Common symptoms include, but are not limited to, fatigue, shortness of breath, chest pain and headache [5, 9–11]. Persistent PASC has not only detrimental effects on one's overall physical and mental health but also on one's functional abilities, social wellbeing and quality of life [7, 12].

Underlying comorbid conditions have been established as a key risk factor contributing to the acquisition and severity of COVID-19 [13, 14]. Therefore, the impact of COVID-19 on at-risk populations such as people living with HIV (PLWH) are a continuing concern. Existing research is equivocal as to whether HIV seropositivity alone is associated with higher disease prevalence and adverse outcomes in both acute COVID-19 infection and PASC [15, 16]. However, PWLH often face overlapping vulnerabilities related to other risk factors (e.g., age, comorbidities) and social determinants of health [17]. For example, there is an overlap between sociodemographic characteristics associated with increased risk of COVID infection severity, and mortality and disproportionate impact of HIV (e.g., race/ethnicity) [18, 19]. Consequently, PWLH, particularly those who are older and/or from historically marginalized communities have higher rates of COVID-19 incidence and hospitalization [16, 20].

Factors contributing to PASC development remain inadequately understood given the relative recent emergence of this syndrome. To-date, certain risk factors for PASC have been identified including female sex, older age, obesity, and tobacco use [1, 21, 22]. However, few studies have focused on race/ethnic and socioeconomic disparities in PASC. While numerous comorbidities and health behaviors have been associated with greater risk and severity of SARS-COV-2 infection (e.g., diabetes, tobacco use, respectively), less is known about how factors are related to PASC [23–25]. Given the substantial overlap between social determinants of health for HIV and for COVID-19, understanding how PASC affects PLWH is critical. The primary aim of this study was to identify the long-term symptoms of COVID-19 reported by a Medicaid managed care population and explore whether HIV serostatus, sociodemographic factors, and comorbidities are associated with COVID-19 symptoms and symptom duration. Based on the emergent literature, we focused on PASC symptoms experienced for six months or longer [4–7]. Given the exploratory nature of this research, we sought to answer the following research questions:

1. Does the prevalence of PASC and intensity (number of PASC symptoms) differ between people with and those without HIV and other sociodemographic characteristics?

2. What is the association between the intensity of PASC with comorbidities and health behaviors (substance use)?

To answer our research questions, we employed a web-based survey, including questions concerning experience of PASC symptoms, HIV serostatus, socio-demographic information, comorbidities and health behaviors, among a Medicaid managed care population who had been previously diagnosed with acute COVID-19 infection according to their health insurance records.

## Methods

### Procedures

This study partnered with a Medicaid Special Needs Plan (SNP) for people with and at risk for HIV. Data was collected using a web-based survey sent to enrolled members of the plan.

Potential participants were identified through insurance claims data if they had received a diagnosis of COVID-19 infection. Eighty percent of the sample pool were HIV-positive. Recruitment occurred between July 2022 and January 2023 via text messaging and mailed flyers. Interested members could access the online Qualtrics survey via a link provided by the recruitment text messages or by a QR code on the mailed recruitment flier. Participants provided web-based informed consent and received a $20 gift card as compensation upon survey completion. Recruitment, informed consent, and survey materials were available in English and Spanish.

The study protocol was approved by the City University of New York Institutional Review Board. Before participants can proceed to the questionaries, they were prompted to review the informed consent page. All participants gave their electronic informed consent.

## Measures

The survey consisted of 50 questions encompassing multiple-choice questions, likert scales, and open-ended responses. Initial sets of questions concerned participants' socio-demographic information, including age, race/ethnicity, gender identity, sex assigned at birth, sexual orientation, educational attainment, income adequacy, and living arrangements. Participants were asked to rate their health and provide details about pre-existing risk factors for COVID-19, such as height and weight for computation of body mass index (BMI), substance use, and comorbidities. For the purpose of this study, we focused on ten comorbidities that had been identified as increasing risk for poor outcomes of COVID-19 infection: diabetes, hypertension, stroke/heart disease, asthma/COPD, cancer, depression/anxiety, fibromyalgia, bone/joint problems, hepatitis C/liver disease, and kidney disease. Additionally, participants were queried about their HIV serostatus. Participants who answered "yes" to being HIV positive were asked to report CD4 t-cell counts (most recent and nadir below 200) and HIV viral load (detectable/undetectable).

The body of the survey obtained information about PASC symptoms and impact adapted from a validated instrument (V. Tran et al., 2021) [26], which presented 49 PASC symptoms:

- General (weight loss, loss of appetite, sweats, fever and chills, hot flashes, fatigue, sleeping more, difficulty sleeping, heat/cold intolerance, changing mood/impact on morale, body aches)

- Thorax (rib cage pain, chest pressure, sharp sudden pain, chest burns, heart beating too fast/slow/irregular, cough)

- Neurological (headache, tremor, dizziness/malaise, word finding problems, brain fog/difficulty concentrating, memory problems, pricking/tingling/creeping feeling on skin, impaired/decreased sense of touch, change/loss of taste, change/loss of smell)

- Digestive (abdominal pain, nausea/vomiting, diarrhea)

- Ear/nose/throat (sore throat/tongue/mouth, trouble swallowing, ear pain, clogged ears, tinnitus, congested/runny nose, sensitivity to sound)

- Eyes (dry eyes, blurry vision, sensitivity to light)

- Musculoskeletal (bone and joint pain, heavy legs/swelling of legs, muscle aches, neck/back/low back pain)

- Blood and lymph node circulation (bulging veins, unexplained bruising, swollen lymph nodes, high or low blood pressure)

- Skin and hair (dry/peeling skin, hair loss, skin rash, discoloration/swelling of hands and feet)
- Urinary and gynecological problems (changes in menstrual cycle, problems with urination, erectile dysfunction)

Participants were asked to indicate which of the identified symptoms had emerged or worsened after their initial COVID-19 diagnosis, along with specifying the duration of each symptom (1 month or less, 2–5 months, 6 months or more). We computed the proportion of participants who experienced one or more symptoms for each time period, as well as the number of COVID symptoms experienced for each level of duration by summing the number of symptoms reported.

## Participants

Any enrolled member of our partnered SNP who was at least 18 years of age and had a documented COVID-19 diagnosis in their medical record was eligible to participate. A total of 153 responses were received, of which 138 respondents answered the question on HIV serostatus and were usable for the current analysis.

## Design and analysis

This study used a cross-sectional survey design to examine the association of PASC symptoms with sociodemographic characteristics, comorbidities, substance use, and HIV serostatus. Data were imported from Qualtrics into SPSS v.27 for cleaning and data analysis. Univariate analyses were conducted to examine measures of central tendency for continuous variables and frequencies and percentages for categorical variables. We conducted bivariate analyses of the associations between sociodemographic and individual COVID symptoms, as well as the number of symptoms reported lasting six months or more. In preparation for the multivariate regression analyses, we computed correlations between sociodemographic factors, comorbidities, substance use, and the number of COVID symptoms experienced at six months or longer.

The OLS regression model was derived based on the literature reporting on factors relating to SARS-COV-2 and PASC (HIV serostatus, sociodemographic, comorbidities, and substance use). Categorical variables with 3 or more categories were dummy-coded prior to analysis. Reference categories for dummy coded variables were as follows: race/ethnicity (non-Hispanic White, Asian, and other race); age group (20 to 39 years); gender identity (cisgender male), and income adequacy (enough money or money not a problem). Independent variables were examined for potential multicollinearity during the initial correlational analysis (i.e., $r > .49$) and by using collinearity diagnostics in the available OLS regression analysis. We used a hierarchical order of variable entry with sociodemographic and HIV serostatus entered in the first block and comorbidities and substance use entered in the second block. The regression was evaluated in terms of the significance of independent variables based on t-tests of regression coefficients and the significant explained variance ($R^2$) for each block of variables in the model.

## Results

### Participants demographics

Of the 138 participants included in the analysis, 77.5% were HIV positive. Participants had a mean age of 45.4 years ($SD = 11.79$) with more than 40% of the sample being 50 years of age and older. Most identified themselves as non-Hispanic Black or Hispanic (43.5% and 39.1%, respectively). Regarding income adequacy, only 13.0% indicated having enough money, while

55.1% reported "just manage to get by," and 31.9% reported inadequate incomes. Cisgender males comprised 45.7% of our sample, 29.7% were cisgender female, and 24.6% were transgender/gender diverse. The sample was relatively split between heterosexual (46.4%) and sexual minority individuals (53.6%). Among the participants, the most frequently reported comorbidity was depression or anxiety (n = 73, 52.9%). Approximately 37.7% (n = 52) the sample reported either asthma/COPD or hypertension as a comorbidity. The most frequently reported used substance was alcohol (n = 54, 39.1%), followed by marijuana (n = 42, 30.4%) and tobacco (n = 39, 28.3%).

## Association of HIV serostatus with sociodemographic factors

To gain a better understanding of our sample, significant associations between demographic factors and HIV serostatus were examined (see Table 1). Participants with HIV were significantly older with 50% being aged 50 and older as compared with 5.6% in the HIV negative group. Among individuals with HIV, a higher proportion identified as non-Hispanic Black (45.8%) compared with those who were HIV-negative (35.5%). In regard to gender identity, among those with HIV, a majority identified as cis-male (54.2%) as compared to 16.1% in the HIV negative group. Similarly, 33.6% of those with HIV identified as cis-female, while only 16.1% of the HIV negative group identified as such. Meanwhile, transgender or gender diverse individuals accounted for only 12.1% of the HIV-positive group but constituted 67.7% of the HIV-negative group. No significant association was observed between HIV serostatus and sexual orientation or income adequacy. In regard to comorbidities, only hypertension was found to be significantly associated with HIV serostatus, with 43.9% of HIV-positive individuals reporting the condition as compared with 16.1% of those who were HIV-negative, which may be related to the greater age of the HIV-positive group. However, HIV-positive participants reported a higher prevalence of the remaining comorbidities compared to those who were HIV-negative although these differences were not statistically significant.

## Overall PASC symptoms prevalence

Out of the 138 participants, 65 (47.1%) reported one or more PASC symptoms that last for 6 months or more. Overall, the top 10 most frequently reported symptoms lasting over 6 months were neck, back, or low back pain (22.5%); brain fog or difficulty concentrating (21.0%); bone or joint pain (20.3%); muscle aches (20.3%); fatigue (19.6%); memory problems (18.8%); body aches (16.7%); difficulty sleeping (15.2%); changing mood/impact on morale (15.2%); problem finding the right words (15.2%); and dry eyes (15.2%).

## HIV serostatus and PASC symptoms

In this study, we examined the association of HIV serostatus with individual PASC symptoms lasting six months or longer. Among participants living with HIV, 52 (48.6%) reported PASC symptoms lasting over 6 months as compared to 12 (41.9%) of those without HIV. However, this difference was not statistically significant [X2(1) = 0.428, p = 0.546].

As detailed in the supplementary material (See S1 File), the most frequently reported symptoms after 6 months mirrored the overall findings, including neck, back, or low back pain (26.2%), bone or joint pain (23.4%), muscle aches (22.4%), brain fog or difficulty concentrating (21.5%), memory problems (19.6%), fatigue (18.7%), difficulty sleeping (17.8%), body aches (16.8%), changing mood/impact on morale (16.8%), dry eyes (16.8%), and sensitivity to light (16.8%). Among the 31 participants without HIV, fatigue (22.6%), brain fog, difficulty concentrating (19.4%), body aches (16.1%), chest pressure (16.1%), problem finding the right words (16.1%), memory problem (16.1%), change/loss of taste (16.1%), muscle aches (12.9%), and

**Table 1. Demographic and comorbid conditions by HIV serostatus.**

| | Total | | HIV Positive | | HIV Negative | | Chi Square |
|---|---|---|---|---|---|---|---|
| | N | % | N | % | N | % | |
| **Age Groups** | | | | | | | $x^2$ (2) = 14.549*** |
| 20 to 39 | 33 | 33.0 | 21 | 25.6 | 12 | 66.7 | |
| 40 to 49 | 25 | 25.0 | 20 | 24.4 | 5 | 27.8 | |
| 50 and older | 42 | 42.0 | 41 | 50.0 | 1 | 5.6 | |
| **Race and Ethnicity** | | | | | | | $x^2$ (2) = 9.155** |
| Non-Hispanic Black | 60 | 43.5 | 49 | 45.8 | 11 | 35.5 | |
| Hispanic | 54 | 39.1 | 45 | 42.1 | 9 | 29.0 | |
| Non-Hispanic White, Asian, Mixed, Other | 24 | 17.4 | 13 | 12.1 | 11 | 35.5 | |
| **Gender Identity** | | | | | | | $x^2$ (2) = 40.266*** |
| Cisgender Male | 63 | 45.7 | 58 | 54.2 | 5 | 16.1 | |
| Cisgender Female | 41 | 29.7 | 36 | 33.6 | 5 | 16.1 | |
| Transgender/Gender Diverse | 34 | 24.6 | 13 | 12.1 | 21 | 67.7 | |
| **Sexual Identity** | | | | | | | $x^2$ (1) = .024 |
| Heterosexual | 64 | 46.4 | 50 | 46.7 | 14 | 45.2 | |
| Queer | 74 | 53.6 | 57 | 53.3 | 17 | 54.8 | |
| **Income Adequacy** | | | | | | | $x^2$ (1) = .821 |
| Not enough money | 44 | 31.9 | 35 | 32.7 | 9 | 29.0 | |
| Just manage to get by | 76 | 55.1 | 59 | 55.1 | 17 | 54.8 | |
| Enough or not a problem | 18 | 13.0 | 13 | 12.1 | 5 | 16.1 | |
| **Comorbidities** | | | | | | | |
| Hypertension | 52 | 37.7 | 47 | 43.9 | 5 | 16.1 | $x^2$ (1) = 7.908** |
| Diabetes | 23 | 16.7 | 21 | 19.6 | 2 | 6.5 | $x^2$ (1) = 3.004 |
| Stroke or Heart Disease | 12 | 8.7 | 11 | 10.3 | 1 | 3.2 | $x^2$ (1) = 1.507 |
| Asthma or COPD | 52 | 37.7 | 43 | 40.2 | 9 | 29.0 | $x^2$ (1) = 1.274 |
| Cancer | 12 | 8.7 | 12 | 11.2 | 0 | 0.0 | $x^2$ (1) = 3.808 |
| Depression or Anxiety | 73 | 52.9 | 58 | 54.2 | 15 | 48.4 | $x^2$ (1) = .327 |
| Fibromyalgia | 4 | 2.9 | 3 | 2.8 | 1 | 3.2 | $x^2$ (1) = .015 |
| Bone or Joint Problems | 28 | 20.3 | 25 | 23.4 | 3 | 9.7 | $x^2$ (1) = 2.784 |
| Hepatitis C/ Liver Disease | 8 | 5.8 | 8 | 7.5 | 0 | 0.0 | $x^2$ (1) = 2.460 |
| Kidney Disease | 9 | 6.5 | 9 | 8.4 | 0 | 0.0 | $x^2$ (1) = 2.789 |

Note. N = 138. HIV Positive n = 107. HIV Negative n = 31.

*p < .05,

**p < .01,

***p < .001

sleeping more (12.9%) emerged as more frequently reported persistent PASC symptoms over 6 months. There were no significant differences in disease prevalence among the most frequently reported symptoms between people living with HIV and those without HIV.

Among the less frequently reported symptoms, there is a significant connection between HIV-positive status and the symptom of pricking, tingling, or creeping feeling on the skin ($x^2$ (2) = 4.096*, p = 0.043). However, among the total 49 PASC symptoms examined, this association stands out as the sole statistically significant finding. Given the well documented link between neuropathy and HIV infection [27], we conclude that this isolated correlation cannot determine the overall association between HIV serostatus and PASC prevalence or intensity.

**Table 2. PASC symptoms by age group.**

| Symptom | Total | | 20 to 39 | | 40 to 49 | | 50 and older | | Chi-Square |
|---|---|---|---|---|---|---|---|---|---|
| | N | % | N | % | N | % | N | % | |
| **Heat/cold intolerance** | | | | | | | | | |
| No or less than five months | 91 | 91.0 | 33 | 100.0 | 25 | 100.0 | 33 | 78.6 | |
| Six months or more | 9 | 9.0 | 0 | 0.0 | 0 | 0.0 | 9 | 21.4 | $x^2 (2) = 13.658***$ |
| **Muscle aches** | | | | | | | | | |
| No or less than five months | 85 | 85.0 | 32 | 97.0 | 21 | 84.0 | 32 | 76.2 | |
| Six months or more | 15 | 15.0 | 1 | 3.0 | 4 | 16.0 | 10 | 23.8 | $x^2 (2) = 6.284*$ |
| **Neck, back, low back pain** | | | | | | | | | |
| No or less than five months | 82 | 82.0 | 31 | 93.9 | 21 | 84.0 | 30 | 71.4 | |
| Six months or more | 18 | 18.0 | 2 | 6.1 | 4 | 16.0 | 12 | 28.6 | $x^2 (2) = 6.435*$ |

Note. N = 100. 20 to 39 n = 33. 40 to 49 n = 25. 50 and older n = 42.

*p < .05,

**p < .01,

***p < .001

## Demographic factors associated with PASC symptoms

We examined the association of demographic factors (age, race/ethnicity, sexual orientation, gender identity, income adequacy) with individual PASC symptoms of six months or greater duration, and found consistent patterns with regard to age and income adequacy. As indicated in Table 2, greater age was associated with an elevated risk of experiencing various PASC symptoms beyond 6 months. Muscle aches were more prevalent among the 50+ age group (23.8%) compared to the 40–49 and 20–39 age brackets (16.0% and 3.0%, respectively). The occurrence of neck, back and low back pain was more pronounced among those over 50 (28.6%) than the younger age groups (16.0% and 6.1%, respectively). Furthermore, age over 50 emerged as a strong predictor of heat/cold intolerance. While none of the younger counterparts reported experiencing heat/cold tolerance, 21.4% participants over 50 reported this symptom.

Regarding income adequacy (see Table 3), individuals with inadequate income were more susceptible to a wide range of symptoms, compared to those who just managed to get by or those with sufficient financial adequacy. These symptoms included sleep difficulty (27.3%, 9.2%, and 11.1%, respectively), rib cage pain (15.9%, 2.6%, and 5.6%, respectively), brain fog (34.1%, 17.1%, and 5.6%, respectively), dry eyes (27.3%, 7.9%, & 16.7%, respectively), muscle aches (34.1, 15.8%, and 5.6%, respectively), neck, back, or low back pain (36.4%, 17.1%, and 11.1%, respectively), and hair loss (20.5%, 9.2%, and 0%, respectively). Additionally, mood changes were strongly associated with income inadequacy (29.5%) compared to just managing to get by (9.2%) or having sufficient income (5.6%). More notably, individuals with inadequate incomes were significantly more likely to experience heat/cold intolerance (25.0% vs 5.3% vs 5.6%) and body aches (36.4% vs 6.6% vs 11.1%), compared to those just managing to get by or with sufficient income.

## Multivariate analysis

We conducted an OLS regression analysis regarding our second research question concerning the association of sociodemographic factors, comorbidities, and health behaviors (e.g., tobacco use) that have been associated with SARS-COV-2 incidence and severity with the number of

**Table 3. PASC symptoms by income adequacy.**

| Symptoms | Total | | Not Enough Money | | Just Manage to Get By | | Enough or Not a Problem | | Chi-Square |
|---|---|---|---|---|---|---|---|---|---|
| | N | % | N | % | N | % | N | % | |
| **Difficulty sleeping** | | | | | | | | | |
| No or less than five months | 117 | 84.8 | 32 | 72.7 | 69 | 90.8 | 16 | 88.9 | |
| Six months or more | 21 | 15.2 | 12 | 27.3 | 7 | 9.2 | 2 | 11.1 | $x^2 (2) = 7.317^*$ |
| **Heat/cold intolerance** | | | | | | | | | |
| No or less than five months | 122 | 88.4 | 33 | 75.0 | 72 | 94.7 | 17 | 94.4 | |
| Six months or more | 16 | 11.6 | 11 | 25.0 | 4 | 5.3 | 1 | 5.6 | $x^2 (2) = 11.327^{***}$ |
| **Changing mood/impact on morale** | | | | | | | | | |
| No or less than five months | 117 | 84.8 | 31 | 70.5 | 69 | 90.8 | 17 | 94.4 | |
| Six months or more | 21 | 15.2 | 13 | 29.5 | 7 | 9.2 | 1 | 5.6 | $x^2 (2) = 10.429^{**}$ |
| **Body aches** | | | | | | | | | |
| No or less than five months | 115 | 83.3 | 28 | 63.6 | 71 | 93.4 | 16 | 88.9 | |
| Six months or more | 23 | 16.7 | 16 | 36.4 | 5 | 6.6 | 2 | 11.1 | $x^2 (2) = 18.259^{***}$ |
| **Rib cage pain** | | | | | | | | | |
| No or less than five months | 128 | 92.8 | 37 | 84.1 | 74 | 97.4 | 17 | 94.4 | |
| Six months or more | 10 | 7.2 | 7 | 15.9 | 2 | 2.6 | 1 | 5.6 | $x^2 (2) = 7.397^*$ |
| **Brain fog, difficulty concentrating** | | | | | | | | | |
| No or less than five months | 109 | 79.0 | 29 | 65.9 | 63 | 82.9 | 17 | 94.4 | |
| Six months or more | 29 | 21.0 | 15 | 34.1 | 13 | 17.1 | 1 | 5.6 | $x^2 (2) = 7.824^*$ |
| **Abdominal pain** | | | | | | | | | |
| No or less than five months | 133 | 96.4 | 41 | 93.2 | 76 | 100.0 | 16 | 88.9 | |
| Six months or more | 5 | 3.6 | 3 | 6.8 | 0 | 0.0 | 2 | 11.1 | $x^2 (2) = 7.034^*$ |
| **Dry eyes** | | | | | | | | | |
| No or less than five months | 117 | 84.8 | 32 | 72.7 | 70 | 92.1 | 15 | 83.3 | |
| Six months or more | 21 | 15.2 | 12 | 27.3 | 6 | 7.9 | 3 | 16.7 | $x^2 (2) = 8.144^*$ |
| **Muscle aches** | | | | | | | | | |
| No or less than five months | 110 | 79.7 | 29 | 65.9 | 64 | 84.2 | 17 | 94.4 | |
| Six months or more | 28 | 20.3 | 15 | 34.1 | 12 | 15.8 | 1 | 5.6 | $x^2 (2) = 8.550^*$ |
| **Neck, back, low back pain** | | | | | | | | | |
| No or less than five months | 107 | 77.5 | 28 | 63.6 | 63 | 82.9 | 16 | 88.9 | |
| Six months or more | 31 | 22.5 | 16 | 36.4 | 13 | 17.1 | 2 | 11.1 | $x^2 (2) = 7.466^*$ |
| **Hair loss** | | | | | | | | | |
| No or less than five months | 122 | 88.4 | 35 | 79.5 | 69 | 90.8 | 18 | 100.0 | |
| Six months or more | 16 | 11.6 | 9 | 20.5 | 7 | 9.2 | 0 | 0.0 | $x^2 (2) = 6.152^*$ |

Note. N = 138. Not enough money, n = 44. Just manage to get by, n = 76. Enough or not a problem, n = 18.

$^*p < .05$,

$^{**}p < .01$,

$^{***}p < .001$

PASC symptoms lasting 6 months or more reported by Medicaid Managed Care enrollees having or being at risk for HIV. We included those who reported no PASC symptoms of six months or more in this analysis. Prior to regression analysis, we examined the bivariate correlations of independent variables with the number of PASC symptoms lasting 6 months or more to develop a parsimonious regression model that would be sufficiently powered given our available sample size. Only variables that were significantly correlated with PASC

**Table 4. Multiple regression analysis on number of long-COVID symptoms at six-months in a Medicaid managed care population.**

| Variable | r | B | SE | β | t | p | $R^2$ Change |
|---|---|---|---|---|---|---|---|
| **Block One:** | | | | | | | |
| Non-Hispanic Black | -.116 | -2.262 | 1.667 | -.144 | -1.357 | .177 | |
| Hispanic | .045 | -1.992 | 1.705 | -.125 | -1.168 | .245 | |
| Inadequate Income | .286*** | 2.946 | 1.252 | .176 | 2.354 | .020 | |
| HIV+ | -.084 | -0.603 | 1.441 | -.032 | -0.419 | .676 | |
| | | | | | | | .103** |
| **Block Two:** | | | | | | | |
| Stroke or Cardiac | .383*** | 7.386 | 2.218 | .267 | 3.329 | .001 | |
| Cancer | .254*** | 5.478 | 2.078 | .198 | 2.636 | .009 | |
| Depression or Anxiety | .213** | 1.888 | 1.170 | .121 | 1.614 | .109 | |
| Fibromyalgia | .240** | 7.409 | 3.546 | .159 | 2.089 | .039 | |
| Bone or Joint Problem | .267*** | 1.952 | 1.596 | .101 | 1.223 | .223 | |
| Use Tobacco | .239** | 2.429 | 1.355 | .140 | 1.793 | .075 | |
| | | | | | | | .238*** |

*Note.* N = 138. Total $R^2$ = .341.

* $p < .05$,

** $p < .01$,

*** $p < .001$

symptoms at 6 months were included in the regression model, with the exception of those we considered to be important control factors based on the literature (i.e., race/ethnicity) and the composition of our sample (i.e., HIV serostatus). Variables of the regression analysis were entered in 2 blocks; 1) sociodemographic factors (race/ethnicity, income inadequacy, and HIV serostatus) and 2) comorbidities and health behaviors (stroke/cardiac condition, cancer, depression/anxiety, fibromyalgia, bone/joint problems, tobacco use) (see Table 4). We included HIV serostatus in the first block since it was a defining characteristic of our sample.

Among the sociodemographic variables in Block One, only reporting inadequate income was significantly associated with the number of PASC symptoms lasting six months or more (β = .176), Sociodemographic variables explained 10% of the variance in 6-months+ PASC symptoms. In Block Two, the comorbidities of stroke/cardiac conditions (β = .267), cancer (β = .198, and fibromyalgia (β = .159) were significantly associated with PASC symptoms lasting 6 months or longer. Depression/anxiety, bone/joint problems, and tobacco use were not significantly associated with the number of PASC symptoms 6-month+ when controlling for sociodemographic factors and other comorbidities. Block Two explained 24% of the variance in the number of PASC symptoms lasting 6 months or more (see Table 4). In summary, the number of long duration PASC symptoms were positively associated with income inadequacy and several comorbidities (cardiac/stroke, cancer, fibromyalgia) among Medicaid managed care enrollees with or at risk for HIV. However, HIV-status per se was not significantly related to the number of PASC symptoms at 6-months or more either in bivariate analysis or when controlling for sociodemographic factors, comorbidities, or health behaviors.

## Discussion

Aligning with national demographic trends concerning individuals at elevated risk for HIV [28], in our study, those living with HIV are more likely to identify as non-Hispanic Black. Reflecting the average age of individuals living with HIV in the U.S., half of our participants

with HIV are over 50. HIV positive status is significantly associated with hypertensions, which may also be attributed to the older profile of this group. It is important to recognize that all the participants in the study belonged to the same Medicaid managed plan for people with or at risk for HIV. The homogeneity of our sample may consequently contribute to the lack of association between HIV serostatus noted in other samples. Nevertheless, this study draws meaningful comparisons and findings in an overall low-income population with similar access to healthcare and benefits.

Based on data from a Medicaid Managed population, this study demonstrated that lower income adequacy, and a range of comorbidities are associated with a higher prevalence of persistent symptoms of PASC at more than 6 months after the initial acute COVID-19 infection.

Most of the common symptoms reported by this population are consistent with past findings, which included musculoskeletal conditions (body ache, low back pain etc.), brain fog, fatigue, sleep difficulties, and neurocognitive issues (memory loss, bone, or joint pain etc.) [5, 8–11].

Despite the statistically significant link between one PASC symptom (pricking, tingling, and creeping feeling on the skin) and HIV serostatus, we did not find a significant association between HIV serostatus and either the likelihood of experiencing one or more PASC symptoms six months after infection nor the number of PASC symptoms at six months. This finding contradicts with Peluso and colleague's study (2022) that measured COVID-19 humoral and cellular response in PWLH and predicted a positive correlation between HIV serostatus and PASC acquisition and persistent symptoms [2]. However, it is important to note that the current evidence on the long-term clinical consequences of COVID-19 among PLWH is still quite limited, leaving this study with very few other research to draw meaningful comparison. Additionally, participants in this study all came from the same Medicaid-managed plan, representing a relatively homogenous population, while the existing few studies demonstrating a significant link between positive HIV serostatus and PASC intensity were conducted in more heterogenous population [29].

Moreover, the lack of association between HIV and PASC is not entirely surprising, as several studies on HIV and the acute COVID infection suggested a similar conclusion, indicating that comorbidities and sociodemographic factors may play a more prominent role in determining COVID-19 outcomes than HIV serostatus in PLWH [15, 17, 30, 31]. This study represents one of the very first few studies that integrated PASC and HIV co-infection, aiming to inform the ongoing exploration of future studies in this area.

Multiple studies have collectively identified older age and female sex as the potential risk factors for PASC and longer disease persistence [1, 21, 22]. A few studies also have revealed that non-Hispanic Black and Hispanics have a higher likelihood of developing PASC and experiencing a wide range of symptoms compared to their non-Hispanic White counterparts [32, 33]. In the present study, while the association between growing age and the heightened risks of certain PASC symptoms is further strengthened, no apparent pattern was observed between race/ethnicity, gender/sex and the prevalence of PASC or the number of symptoms reported. A plausible explanation to such a difference could be the uniqueness of the study sample that primarily consisted of a Medicaid managed population, in other words, an overall low-income population. In the current sample, there was no significant association between race/ethnicity and income adequacy, therefore further suggesting the possibility that income adequacy could function as an independent risk factor for PASC.

The current study has several limitations including a limited sample size drawn from a distinct clinical population and the use of retrospective self-report data on PASC symptoms, thus, findings may not be generalizable to populations that are not Medicaid enrollees or who do not have or are not at risk for HIV. In addition, there is an imbalance in numbers between

people living with HIV and those without, with people living with HIV constituting the majority. This discrepancy may account for the lack of significant findings regarding the correlation between PASC prevalence and HIV serostatus. However, the use of such a homogeneous population did allow us to compare individuals with similar access to healthcare and healthcare benefits, and our multivariate analysis explained over one-third for the variance in long-duration PASC symptoms and were consistent with other reported findings. Further, we observed adequate statistical power in our regression analysis based on the proportion of variance explained in our model ($R^2$ = .341).

Although empirical findings on the socioeconomic disparity in PASC are scarce, this study's findings aligned with two existing studies that described the increased risk of PASC acquisition and severity among people who are economically disadvantaged [33, 34]. It is worth noting that these two large community-based studies encompassed participants with a wider range of socioeconomic statuses, whereas all of the participants in this study were already low-income. The study findings illustrate the deleterious impact of financial strain even among those with limited financial means in the development of PASC. These findings highlight the need for targeted interventions and support for low-income individuals who are disproportionately affected by PASC, particularly for those who are struggling financially.

## Public health implications

Income inadequacy is a known social determinant of health and was the only demographic characteristic consistently associated with PASC symptoms experienced for six months or longer. Access to healthcare can be directly linked to socioeconomic status, however, the study sample were all Medicaid managed care enrollees and could be assumed to have the same level of access to healthcare. But an additional factor is that those who struggle financially often work multiple jobs and may have difficulty accessing health care and treatments that can limit the long-term impact of SARS-COV-2 and other diseases (e.g., Paxlovid) due to time constraints. COVID-19 has had a disproportionate impact on disadvantaged communities and including those of low socioeconomic position. Acknowledging the potential long-term physical, social, and economic effects of COVID-19 is a crucial first step in improving the public health response to this disease and other health inequities.

## Supporting information

**S1 File. All 49 PASC symptoms by HIV serostatus.**
(DOCX)

## Acknowledgments

This work would not have been possible without the participants of this study. Their insights and openness to share their stories have been invaluable. Furthermore, we want to express our deep gratitude to our colleagues at Amida Care and its Member Advisory Committee for their support in recruitment and operational support.

## Author Contributions

**Conceptualization:** Carey Brandenburg, Richard Overbey, Jerome Ernst, Mark Brennan-Ing.

**Data curation:** Yiyi Wu, Eleni Mattas.

**Formal analysis:** Yiyi Wu, Eleni Mattas, Mark Brennan-Ing.

**Funding acquisition:** Carey Brandenburg, Ethan Fusaris, Richard Overbey, Jerome Ernst, Mark Brennan-Ing.

**Investigation:** Jerome Ernst.

**Methodology:** Yiyi Wu, Mark Brennan-Ing.

**Project administration:** Yiyi Wu.

**Resources:** Carey Brandenburg.

**Supervision:** Ethan Fusaris, Jerome Ernst, Mark Brennan-Ing.

**Writing – original draft:** Yiyi Wu, Eleni Mattas, Mark Brennan-Ing.

**Writing – review & editing:** Yiyi Wu, Carey Brandenburg, Jerome Ernst, Mark Brennan-Ing.

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
