## [Decision Letter · Decision Letter 0]

6 Mar 2024

PONE-D-23-39018The association of sociodemographic characteristics and comorbidities with post-acute sequelae of SARS-CoV-2 in a Medicaid managed care population with and without HIVPLOS ONE

Dear Dr. Wu,

Thank you for submitting your manuscript to PLOS ONE. After careful consideration, we feel that it has merit but does not fully meet PLOS ONE’s publication criteria as it currently stands. Therefore, we invite you to submit a revised version of the manuscript that addresses the points raised during the review process.

We look forward to receiving your revised manuscript.

Kind regards,

Shuai Ren

Academic Editor

PLOS ONE

 [copy in funding statement].  

3. In the online submission form, you indicated that [Data are available upon reasonable request to the authors.]. 

Reviewers' comments:

Reviewer's Responses to Questions

**Comments to the Author**

1. Is the manuscript technically sound, and do the data support the conclusions?

Reviewer #1: Partly

2. Has the statistical analysis been performed appropriately and rigorously? 

Reviewer #1: No

3. Have the authors made all data underlying the findings in their manuscript fully available?

Reviewer #1: No

4. Is the manuscript presented in an intelligible fashion and written in standard English?

Reviewer #1: Yes

5. Review Comments to the Author

Reviewer #1: PLOS ONE

PONE-D-23-39018

Original research

The association of sociodemographic characteristics and comorbidities with post-acute sequelae of SARS-CoV-2 in a Medicaid managed care population with and without HIV.

Wu et al. investigate the prevalence of post-acute sequelae of SARS-CoV-2 infection (PASC) between people with and those without HIV from a Medicaid Managed population using a web-based survey. Authors also evaluate the nature of PASC symptoms in the two populations as well as the influence of sociodemographic factors, comorbidities and health behaviors in the number of PASC symptoms.

The study unveils relevant data on the lack of significant association between HIV status and PASC.

Authors should better clarify several methodological issues, before the manuscript can be considered for publication:

1. Structured abstract section should clearly describe the results according to the study research questions

2. Authors claim the study is a cohort-based study (page 16, line 295). Please, explain

3. Introduction section: For such relevant research question, is the methodology appropriate (Page 3, lines 82-83)? Please, explain

4. Methods section: How accurate is the methodology used to identify people at risk for or living with HIV: “Participants who answered “yes” to being with HIV positive were asked to report CD4 t-cell counts and HIV viral load (detectable / undetectable)” (Page 4, lines 113-115). Please, comment.

5. Methods section: How was the diagnosis of COVID-19 confirmed? What does 6 months or more refers to (Page 9, line 213)? Authors should inform the time since the acute SARS-CoV-2 infection.

6. Results section: Percentage of included people reporting HIV and non-HIV is unbalanced. Could this finding affect the study results? Please, clarify and discuss.

7. Results section: Table 1 introduces the demographic and comorbid conditions by HIV status: According to age count there is information of only 100 participants (Page 8, line 2090, not 153 (Page 1, line 320, nor of the 142 participants included in the study 9page 7, line 176). For race and ethnicity, gender identify, sexual identify and income adequacy authors provide data from 138 participants (Page 8, line 209). Please, clarify.

8. Results section: Prevalence of long COVID symptoms only informs the total prevalence, and NOT the difference found in people with and those without HIV (Page 9, lines 212-218).

9. Results section: Table 2 introduces the percentage of PASC symptoms by age group, but not by HIV status (Page 10, line 235). Table 3 introduces the percentage of PASC symptoms by income adequacy, but not by HIV status (Page 11, line 250).

10. Results section: Table 4 presents the multiple regression analysis on the number of Long-COVID symptoms at six-months in a Medicaid Managed Care Population: is data derived from the HIV population only? Is data derived from people who represented symptoms only at six-months? Please, kindly clarify.

11. Discussion section should include relevant comments regarding study findings based on HIV status. How did sociodemographic characteristics and comorbidities differ in people with and without HIV?

12. Long COVID (Page 9, line 212; Page 14, line 269). Authors should consider harmonizing terminology.

6. PLOS authors have the option to publish the peer review history of their article (what does this mean?). If published, this will include your full peer review and any attached files.

Reviewer #1: No

---

## [Author Response · Author response to Decision Letter 0]

11 Apr 2024

Dear Editor and Reviewer at PlosOne,

Thank you so much for the opportunity to revise our manuscript “The association of sociodemographic characteristics and comorbidities with post-acute sequelae of SARS-CoV-2 in a Medicaid managed care population with and without HIV.” We appreciate your very insightful feedback and we have made improvements to our manuscript. We have addressed each of your comment as listed below. 

Editor’s Comment 1: Please ensure that your manuscript meets PLOS ONE's style requirements, including those for file naming. The PLOS ONE style templates can be found at 

Response: Thank you so much for your comment! We have made several formatting adjustments based on the guidelines. We hope that these changes adequately resolve all formatting-related issues. 

Editor’s Comment 2: Thank you for stating the following financial disclosure: [copy in funding statement]. Please state what role the funders took in the study. If the funders had no role, please state: ""The funders had no role in study design, data collection and analysis, decision to publish, or preparation of the manuscript. “If this statement is not correct you must amend it as needed. Please include this amended Role of Funder statement in your cover letter; we will change the online submission form on your behalf.

Response: Thank you so much for seeking clarification! In our study, the funder played a role in consulting our study design and data collection. Additionally, in accordance with our contract agreement, the funder reviewed the manuscript before its submission.

Editor’s Comment 3: In the online submission form, you indicated that [Data are available upon reasonable request to the authors.]. All PLOS journals now require all data underlying the findings described in their manuscript to be freely available to other researchers, either 1. In a public repository, 2. Within the manuscript itself, or 3. Uploaded as supplementary information. This policy applies to all data except where public deposition would breach compliance with the protocol approved by your research ethics board. If your data cannot be made publicly available for ethical or legal reasons (e.g., public availability would compromise patient privacy), please explain your reasons on resubmission and your exemption request will be escalated for approval. 

Response: The data collected for this study are not publicly available, nor are they available on request, as it would constitute a violation of the Institutional Review Board (IRB) protocol. Additionally, as part of the informed consent agreement, participants were assured that their data would not be shared with researchers beyond our institution or used for future research purposes. We are committed to upholding these ethical standards to protect participant confidentiality and privacy.

Editor’s Comment 4: We note that you have indicated that there are restrictions to data sharing for this study. PLOS only allows data to be available upon request if there are legal or ethical restrictions on sharing data publicly. For more information on unacceptable data access restrictions, please see http://journals.plos.org/plosone/s/data-availability#loc-unacceptable-data-access-restrictions. Before we proceed with your manuscript, please address the following prompts: a) If there are ethical or legal restrictions on sharing a de-identified data set, please explain them in detail (e.g., data contain potentially identifying or sensitive patient information, data are owned by a third-party organization, etc.) and who has imposed them (e.g., a Research Ethics Committee or Institutional Review Board, etc.). Please also provide contact information for a data access committee, ethics committee, or other institutional body to which data requests may be sent. b) If there are no restrictions, please upload the minimal anonymized data set necessary to replicate your study findings to a stable, public repository and provide us with the relevant URLs, DOIs, or accession numbers. For a list of recommended repositories, please see

https://journals.plos.org/plosone/s/recommended-repositories. You also have the option of uploading the data as Supporting Information files, but we would recommend depositing data directly to a data repository if possible. We will update your Data Availability statement on your behalf to reflect the information you provide.

Response: Thank you so much for your comment. As indicated in our last response, we have revised our data availability statement to reflect on the ethical and legal constraints imposed by the Institutional Review Board at the City University of New York. The new data availability statement is presented as below: 

“The data collected for this study are not publicly available, nor are they available on request, as it would constitute a violation of the Institutional Review Board (IRB) protocol. Additionally, as part of the informed consent agreement, participants were assured that their data would not be shared with researchers beyond our institution or used for future research purposes. We are committed to upholding these ethical standards to protect participant confidentiality and privacy.”

Editor’s Comment 5: Please include your full ethics statement in the ‘Methods’ section of your manuscript file. In your statement, please include the full name of the IRB or ethics committee who approved or waived your study, as well as whether or not you obtained informed written or verbal consent. If consent was waived for your study, please include this information in your statement as well. 

Response: Thank you so much for your comment. I have inserted the ethics statement in the methods section with detailed description of informed consent obtainment process. 

 “The study protocol was approved by the City University of New York Institutional Review Board. Before participants can proceed to the questionaries, they were prompted to review the informed consent page. All participants gave their informed consent digitally.” [LN102-104 in file “Manuscript”]

Reviewer’s comment 1: Structured abstract section should clearly describe the results according to the study research questions.

Response: Thank you so much for your comment! We have revised the abstract as instructed. Now the response to the research questions is more clearly listed. 

Reviewer’s comment 2: Authors claim the study is a cohort-based study (page 16, line 295). Please, explain.

Response: Thank you for pointing this out! We initially included the “cohort-based” to indicate the cohort of the care population. However, we realized that it may have caused confusion. We have deleted the term “cohort-based” throughout our study.

Reviewer’s comment 3: Introduction section: For such relevant research question, is the methodology appropriate (Page 3, lines 82-83)? Please, explain.

Response: Thank you so much for your comment! We have modified the languages in the manuscript to make it more clearly align with the research questions. [LN 82-89 in file “Manuscript”]

Reviewer’s comment 4: Methods section: How accurate is the methodology used to identify people at risk for or living with HIV: “Participants who answered “yes” to being with HIV positive were asked to report CD4 t-cell counts and HIV viral load (detectable / undetectable)” (Page 4, lines 113-115). Please, comment.

Response: Thank you so much for your request of clarification! The HIV status in this study is self-reported. We understand that self-report status can be inaccurate in some cases. However, as we clearly stated in the manuscripts that all participants are from a Medicaid managed plan particularly for people live with or at risk for HIV. Please let us know if this addresses your concerns.

Reviewer’s comment 5: Methods section: How was the diagnosis of COVID-19 confirmed? What does 6 months or more refers to (Page 9, line 213)? Authors should inform the time since the acute SARS-CoV-2 infection. 

Response: Thank you so much for your comment! We have clarified that we identified this population through participants’ medical record [LN 89; LN 151]. In other words, only those who “had a documented COVID-19 diagnosis in their medical record was eligible to participate. [LN 151]” “6 months or more” refers to symptoms that last 6 months or more. Now, we have modified that language [LN 216].

Reviewer’s comment 6: Results section: Percentage of included people reporting HIV and non-HIV is unbalanced. Could this finding affect the study results? Please, clarify and discuss.

Response: Thank you for pointing out this issue! We are aware of the imbalance between people reporting HIV and non-HIV, which could affect the study result. Now we have included a few sentences in our discussion in the revised manuscript [LN 367-370].

Reviewer’s comment 7: Results section: Table 1 introduces the demographic and comorbid conditions by HIV status: According to age count there is information of only 100 participants (Page 8, line 2090, not 153 (Page 1, line 320, nor of the 142 participants included in the study 9page 7, line 176). For race and ethnicity, gender identify, sexual identify and income adequacy authors provide data from 138 participants (Page 8, line 209). Please, clarify.

Response: Thank you for pointing this issue out! Such a difference in the number of participants is due to the fact that it is a survey study, and participants were not obligated to answer every single question. Nevertheless, as you insightfully pointed out, such an inconsistency may have caused confusion. Now, we have adjusted the valid responses as the 138 participants who had responded to the HIV status question. In response, we have adjusted the tables, demographic information, as well as the analysis accordingly. 

Reviewer’s comment 8: Results section: Prevalence of long COVID symptoms only informs the total prevalence, and NOT the difference found in people with and those without HIV (Page 9, lines 212-218).

Response: Thank you for your insights! As demonstrated in the study, HIV serostatus is not significantly associated with PASC symptoms. To make this finding more explicit and clearer, we have now dedicated a whole sub-section titled “HIV serostatus and PASC symptoms” with an addition of a supplementary material demonstrating the prevalence of 49 PASC symptoms by HIV status. 

Reviewer’s comment 9: Results section: Table 2 introduces the percentage of PASC symptoms by age group, but not by HIV status (Page 10, line 235). Table 3 introduces the percentage of PASC symptoms by income adequacy, but not by HIV status (Page 11, line 250).

Response: Thank you for your insights! We have now dedicated a whole sub-section in findings titled “HIV serostatus and PASC symptoms” and added a supplementary material demonstrating the prevalence of 49 PASC symptoms by HIV serostatus. 

Reviewer’s comment 10: Results section: Table 4 presents the multiple regression analysis on the number of Long-COVID symptoms at six-months in a Medicaid Managed Care Population: is data derived from the HIV population only? Is data derived from people who represented symptoms only at six-months? Please, kindly clarify.

Response: Thank you so much for your comment! The table derived from people who represented symptoms only at six months. We have now included the clarification in the manuscript! [LN 280-282]

Reviewer’s comment 11: Discussion section should include relevant comments regarding study findings based on HIV status. How did sociodemographic characteristics and comorbidities differ in people with and without HIV?

Response: Thank you so much for your comment! We have now included a discussion regarding demographic findings based on HIV status in the discussion section [LN312-320].

Reviewer’s comment 12: Long COVID (Page 9, line 212; Page 14, line 269). Authors should consider harmonizing terminology.

Response: Thank you so much for your comment! We have revised the terminology and made sure it is consistent throughout the manuscript.

---

## [Decision Letter · Decision Letter 1]

17 Jun 2024

The association of sociodemographic characteristics and comorbidities with post-acute sequelae of SARS-CoV-2 in a Medicaid managed care population with and without HIV

PONE-D-23-39018R1

Dear Dr. Wu,

We’re pleased to inform you that your manuscript has been judged scientifically suitable for publication and will be formally accepted for publication once it meets all outstanding technical requirements.

Kind regards,

Shuai Ren

Academic Editor

PLOS ONE

Additional Editor Comments (optional):

Reviewers' comments:

Reviewer's Responses to Questions

**Comments to the Author**

1. If the authors have adequately addressed your comments raised in a previous round of review and you feel that this manuscript is now acceptable for publication, you may indicate that here to bypass the “Comments to the Author” section, enter your conflict of interest statement in the “Confidential to Editor” section, and submit your "Accept" recommendation.

Reviewer #2: All comments have been addressed

2. Is the manuscript technically sound, and do the data support the conclusions?

Reviewer #2: Yes

3. Has the statistical analysis been performed appropriately and rigorously? 

Reviewer #2: Yes

4. Have the authors made all data underlying the findings in their manuscript fully available?

Reviewer #2: Yes

5. Is the manuscript presented in an intelligible fashion and written in standard English?

Reviewer #2: Yes

6. Review Comments to the Author

Reviewer #2: The authors have addressed my prior concerns. I do have a few minor suggestions. First, line 122 has a misplaced period (.). Two, SARS-CoV-2 and SARS-COV-2 are used in the title and text. SARS-CoV-2 is the standard notation for the virus. Three, another potential consideration for the discussion is that symptoms may vary based on strain and inoculum, and some evidence suggests that ART or PrEP may have an effect on SARS-CoV-2 related morbidity and mortality.

7. PLOS authors have the option to publish the peer review history of their article (what does this mean?). If published, this will include your full peer review and any attached files.

Reviewer #2: No

---

## [Editor Report · Acceptance letter]

20 Jun 2024

PONE-D-23-39018R1 

PLOS ONE

Dear Dr. Wu, 

I'm pleased to inform you that your manuscript has been deemed suitable for publication in PLOS ONE. Congratulations! Your manuscript is now being handed over to our production team.

Kind regards, 

on behalf of

Dr. Shuai Ren 

Academic Editor

PLOS ONE